# A Joint Power, Delay and Rate Optimization Model for Secondary Users in Cognitive Radio Sensor Networks

**DOI:** 10.3390/s20174907

**Published:** 2020-08-31

**Authors:** Bernard Ssajjabbi Muwonge, Tingrui Pei, Julianne Sansa Otim, Fred Mayambala

**Affiliations:** 1School of Computer Science, Xiangtan University, Xiangtan 411105, China; bmuwonge@cis.mak.ac.ug; 2Key Laboratory of Hunan Province for Internet of Things and Information Security, Xiangtan 411105, China; 3Department of Networks, College of Computing & I.S, Makerere University, Kampala 7062, Uganda; sansa@cis.mak.ac.ug; 4Department of Mathematics, Makerere University, Kampala 7062, Uganda; fmayambala@cns.mak.ac.ug

**Keywords:** network resource optimization, Wireless Sensor Networks (WSNs), Cognitive Radio Wireless Sensor Networks (CR-WSNs), primary user (PU), secondary user (SU), transmission power, transmission rate, Transmission delay, Bit Error Rate, Signal-to-Interference Noise Ratio

## Abstract

To maximize the limited spectrum among primary users and cognitive Internet of Things (IoT) users as we save the limited power and energy resources available, there is a need to optimize network resources. Whereas it is quite complex to study the impact of transmission rate, transmission power or transmission delay alone, the complexity is aggravated by the simultaneous consideration of all these three variables jointly in addition to a channel selection variable, since it creates a non-convex problem. Our objective is to jointly optimize the three major variables; transmission power, rate and delay under constraints of Bit Error Rate (BER), interference and other channel limitations. We analyze how total power, rate and delay vary with packet size, network size, BER and interference. The resulting problem is solved using a branch-and-cut polyhedral approach. For simulation of results, we use MATLAB together with the state-of-the-art BARON software. It is observed that an increase in packet size generally leads to an increase in total rate, total power and total transmission delay. It is also observed that increasing the number of secondary users on the channel generally leads to an increased power, delay and rate.

## 1. Introduction

The transmission medium in Wireless Sensor Networks (WSNs) is under-utilized due to the bursty traffic resulting from the network’s event-driven communication [1]. The scarce, yet underutilized resources must therefore be efficiently allocated for better utilization. Unlike WSNs, cognitive radio sensor nodes are intelligent and can access multiple redundant channels. This enables them address some challenges of WSNs that traditionally employ fixed spectrum allocation and are constrained in communication resources and processing capacity. Transmission is made more reliable since they reduce congestion and excessive packet loss [2]. “Cognitive Radio Sensor Networks (CRSNs) are the state-of-the-art communication paradigm for power constrained short range data communication” [3]. To practically realize CRSNs, the optimum packet size must be addressed since most of the optimum packet sizes available are meant for WSNs and/or CRNs and are not optimal for CRSNs [4]. CRSNs are one of the possible technologies implemented for the Internet of Things (IoT) and other future machine-to-machine-based applications most of which are delay sensitive and power constrained [3]. The optimal packet size of CRSNs for maximizing energy efficiency is studied in [4]. The conditions of maintaining allowable licensed primary user (PU) interference level while achieving reliable event detection, are also enforced. In variable conditions of the channel, shorter packets are more optimal and reduce the PU interference though there’s energy wastage due to wide overhead caused by headers and trailers. Conversely, the increase in packet size enhances the throughput and spectrum usage for secondary users (SUs), while enhancing packet loss probability in similar channel conditions [4].

According to [5], data packet size and transmission power level are very important factors in improving energy efficiency and network lifetime for WSNs. To decrease the total effect of Bit Erorr Rate (BER) on loss of packets, the packets should be of small sizes although this is achieved with fragmentation into more data packets leading to increase in energy dissipation. Therefore, data packet size is a very important parameter of power and energy efficiency for improving the lifetime of a Cognitive Radio (CR)-WSN. In addition, to reduce the packet loss probability, there is need to increase the transmission power. There are also a number of applications that are delay intolerant [5]. We aim at minimizing the transmission delay as one of the objectives in our optimization problem. An increase in throughput is realized when the packet size is fixed compared to the exponential distribution of packet size in Cognitive Radio Networks [6]. These do not look at power and energy efficiency and reliability in event detection, yet these are key challenges in sensor networks. In terms of delay and energy efficiency, transmitting data with varying packet sizes rather than fixed data packet length dependent network conditions is advantageous to some extent. When network conditions are bad, re-transmission energy overhead is increased by the big size of packets due to the likely packet collisions. Very small packet size enhances delay overhead emanating from transmission of overhead header, trailer and additional redundant bits save for information bits [3].

Whereas power and energy efficiency are very important in resource-constrained sensor devices, it is not addressed by Cognitive Radio Networks (CRNs). Many currently available protocols from physical layer to upper layers run short of addressing the joint CRN and WSN software and hardware needs and thus may not be applicable in CR-WSNs [1]. We thus need to design power saving, energy-efficient CR-WSN algorithms that are sensitive to the resource limitations of the CR-WSNs to enhance Quality of Service (QoS) in the context of the Internet of Things (IoT), while saving power during transmission. We jointly optimize transmission power, delay and rate with QoS constraints like: interference, delay, BER, Signal-to-Noise-Ratio (SNR) and Signal-to-Interference-Noise-Ratio (SINR). We vary network size and packet size while changing transmission power, delay and rate parameters to sustain QoS constraints. Packet size is varied for; (i) energy efficiency for improved network lifetime due to reduced consumption, (ii) PU interference reduction, (iii) transmission efficiency due to reduced collision between PUs and SUs, (iv) reliability in detection of events. Information carried by a signal during a transmission depends on packet size. Variation of these parameters with network size is also studied.

### Problem Definition

A cross-layer algorithm with optimal channel selection for power optimization (at the physical layer), rate allocation (transport layer) and delay minimization is designed with QoS constraints. QoS is defined by PU interference, BER, SNR, SINR, delay and rate requirements. We employ a multi-objective optimization function with these constraints to solve the resource allocation problem. We maximize transmission rate, minimize transmission power and minimize transmission delay in a constrained environment. We propose a power/energy efficient algorithm with optimum channel selection meant to jointly optimize transmission power, transmission delay and the transmission rate. This leads to power/energy efficiency; improves network QoS by regulating interference and BER; and reduces power wasted by individual nodes during spectrum sensing. To maximize transmission rate, we enable the SU access more than one channel at a time. Since we jointly optimize transmission power, delay and rate with controlled channel selection, this translates to multi-objective optimization. To solve this multi-objective optimization problem, we combine the multiple objectives into a single weighted objective function solved as a non-linear mixed binary optimization problem. We then employ a branch-and-cut algorithm that uses polyhedral cutting planes, to solve the resulting optimization problem. This is an exact optimization technique that guarantees an optimal solution, whenever it exists. The resulting optimal solution then enables us to investigate the rate-power-delay trade-offs by tuning the priority weights associated with each objective. Thus, we study the optimal values against the BER, interference, packet size and network size.

Earlier works have concentrated on bi-objective models. They minimize the power and delay [7] or minimize the power and maximize the transmission rate [1,8,9], but not all of them simultaneously. The authors in [1,9] design a QoS-constrained bi-objective optimization model in order to minimize transmission power and maximize rate. As we design a QoS-constrained multi-objective optimization problem, we minimize the power consumption and transmission delay and maximize transmission rate as we observe their variation with network size and packet size. This is done amidst QoS constraints to improve the user-perceived QoS, which to the best of our knowledge, no prior work has been done combining all these three objectives jointly. Most of the previous works focus on either minimizing power or energy without considering the delay minimization [10] or on delay minimization ignoring power minimization and also do not consider Cognitive Radio Sensor Networks [11].

The rest of this paper is structured as follows: In Section 2, we review the related work; in Section 3, we handle the network model that is used in the study; in Section 4, we formulate the problem whereby we deal with the QoS constraints, that is BER and PU interference, and also we introduce multi-objective optimization for rate, power and delay; in Section 5, we provide the results of our model and the conclusions and recommendations for future work are handled in Section 6.

## 2. Related Literature

Due to the scarcity of spectrum resulting from the proliferation of wireless devices and services, dynamic spectrum access is becoming popular in WSNs since it also provides spectrum efficient communication for the WSNs. Factors like transmission power, fading and interference with licensed users affect the communication between nodes in a CRSN. They also hinder the data transmission among the energy constrained CRS nodes. Hence, there is need for an adaptive energy-efficient optimization scheme which takes into account the varying environment conditions. One of the techniques that has been used in the literature involves the use of optimization of one of the factors, while maintaining an acceptable level (called a constraint) on the other factors. This motivated Jamal et al. [12] to propose a dynamic packet size optimization scheme (DyPSOCS) for CRSN. They employ a constrained Markov decision process (CMDP) to solve the optimization problem with Quality of Service (QoS) constraints. Their scheme improves the QoS performance and energy efficiency but does not support re-transmissions and multiple data transmission per node per cycle. An optimization problem to optimally allocate power in fading channels of a CRN are considered in [13]. Under different channel fading constraints and power constraints, the aim of [13] is to determine an optimal power allocation of SUs that give Ergodic and outage capacities. With a constraint on transmission rate and BER, a model that minimizes power consumption is also developed in [14]. The resulting optimization model in [14] is solved using game theory. It is shown that the Nash equilibrium of the resulting problem is a Pareto-optimal point. An optimization model that determines the optimal packet size of a CRSN, under different QoS constraints is developed by Oto et al. [4]. The authors further show that the obtained packet size also maximizes energy efficiency of the system. Heuristic algorithms to solve the devised model in [4] are handled in [3]. Another power optimization model that maximizes energy efficiency of SUs is handled in [15]. The model is constructed with SINR constraints for both PUs and SUs. When average power constraints and rate constraints are included into the power allocation model, the problem is extensively studied in [16]. Transmission power optimization through a convex design using lower SINR as an SU QoS constraint is studied in [17]. Here, SU power allocation is done through a distributed sub-optimal joint coordination and power control scheme. An optimization model to maximize the transmission rate of SUs in a QoS constrained environment is handled in [17]. The resulting non-linear optimization problem is solved using geometric programming techniques. The main shortfall of single objective optimization is the fact that only one factor is optimized in the entire system. This paper considers a multi-objective optimization problem.

The study of single objective optimization models for the various network variables like power, rate, energy, etc., leads to the obvious question of; *what happens if the variables are jointly optimized?* The answer to this question then gave birth to the many multi-objective optimization models that exist in the literature. A joint power and transmission delay optimization for green wireless access networks is developed in [18,19] as a bi-objective model. The problem is formulated as a mixed integer linear programming problem. However, these make no attempt to include Qos constraints in their model. With delay used as a constraint, a joint power and rate optimization model is developed by [20]. The model is formulated as a dynamic programming problem and solved using three problem-dependent heuristic algorithms. A joint optimization of power and rate with QoS constraints (in terms of minimum SINR and transmission rate and PU interference), is considered by [21]. The authors in [9] design a distributed optimization model for joint determination of optimum power distribution and rate for SUs on a channel to sustain QoS measured using BER and minimum rate requirement. The authors in [22] analyze power and rate adaptation approach for SU capacity optimization constrained by PU interference and BER. The possibility of having many SUs sharing a channel is however ignored. A game theory approach to joint optimization of power and rate under Qos constraints is handled by [23]. The authors demonstrate that the Nash equilibrium points of the joint power and control game is actually Pareto optimal. A pricing scheme to reach the Nash equilibrium point, and hence a Pareto optimal point, is devised and used in [8]. A joint optimization problem for frequency, power and rate allocation in broadband cognitive OFDMA networks is studied in [24]. The resulting non-convex problem is solved using a greedy algorithm and a Lagrangian relaxation algorithm. The authors in [25] adopt a cross-layer strategy that jointly considers routing and MAC layer joint optimization. At the routing layer, they balance traffic by showing that sending traffic generated by each node through multiple paths significantly enables energy conservation. At the MAC layer, they assign the retry limit for retransmissions over each wireless channel. This achieves further energy conservation and improves network lifetime via load balancing. Both [26,27] explore joint allocation of channel and transmission power but ignoring QoS requirements. Further more, the possibility of many SUs transmitting simultaneously across the channel is ignored in [26]. A joint optimization model for power and packet size in wireless sensor networks is studied in [5]. Numerical results for the resulting mixed integer programming problem are extensively explored to show the strength of joint optimization. The problem of energy optimization is re-written as a bi-objective Knapsack problem, whose solution is provided in the paper. For all the factors that have been jointly optimized, non of the earlier works studies the joint multi-objective of power, rate and delay, all considered under different QoS constraints.

There is also a very rich literature on the solution methods that have been used to solve the resulting network optimization problems. The methods range from heuristics, relaxation algorithms and exact algorithms like branch and bound. Xie R, et al. [28] propose a dynamic resource allocation problem for heterogeneous services in CRNs with imperfect channel sensing. They formulate the power and channel allocation problem as a mixed integer programming problem under constraints with the objective of maximizing total capacity of CRNs. For channel sensing, they employ the cooperatively centralized sensing method where SUs sense channels and send sensing information to the secondary Base Station (BS). This sensing method needs backbone infrastructure and has a higher cost. For a large number of users, the bandwidth for reporting is enormous. Most methods employed for spectrum sensing such as energy detection, matched filter detection and cyclo-stationary feature detection also need enormous energy for algorithmic computations. Jamal A, et al. [29] propose an energy efficient and spectrum aware multi-channel MAC protocol for cognitive radio enabled sensor networks. This is a spectrum aware asynchronous duty cycle approach that caters for the requirements of both the CRNs and WSNs. Their scheme employs an asynchronous duty cycle approach for channel acquisition and data transmission [29]. It is a two-dimensional Markov model for changing channel conditions and queues of duty-cycled nodes with a fixed cycle length. Bai Y, et al. [30] propose an energy optimization approach (EOA) for WSNs that exploits the cross-layer principle taking into account the physical layer (transmission power), MAC layer (duty-cycling) and network layer (routing protocol). EOA chooses the optimum routing path using physical layer’s transmission power as metric and determines the MAC layer duty cycle using the routing information of the network layer. The protocol is however designed for WSNs and thus not suitable for CRSNs. Authors in [31] propose a dynamic resource allocation and priority-based scheduling scheme for heterogeneous services in CRNs. They consider a network with heterogeneous SUs in four categories. These include SUs with Minimum Rate Guarantee, SUs with Minimum Delay Guarantee, SUs with Minimum Rate and Delay Guarantee and SUs with Best Effort Service. To reduce the complexity, they assume an equal number of heterogeneous SUs in each of these categories [31]. It is practically impossible to have exactly the same number of clients subscribing to different services at a time. Choosing a channel basing on adaptive task allocation in insect colonies has been done in [32] by use of a biologically motivated spectrum allocation scheme. In addition, for rate optimization, another dual decomposition theory-based distributed system for SUs is developed in [33] for power allocation across sub-carriers in a QoS constrained environment. This is achieved by considering a system model of one CRN alongside multi-cell primary radio networks (PRNs). Each of these works is under multiple frequency sub-bands in each of which many sub-channels are used by the CRN.

## 3. Network Model

We consider a Cognitive Radio Sensor Network (CRSN) with *K* nodes and *I* transmission channels. Each channel can be used for transmission by at most one first-priority user, called a primary user (PU). If a channel is sensed to be free of a PU, an unlicensed user, called a secondary user (SU) can be allowed to transmit on that channel. This is called overlay spectrum sharing approach. The total number of SUs on the network is *K*.

### 3.1. Network Variables

In this paper, we consider four different variables that affect the network system.

(i)Rate: This is the number of bits of a packet that are transmitted per second. The rate of transmission of data for *k*th SU on the *i*th channel shall be denoted as $R_{k}^{i}.$ For ease of notation, we shall drop the subscript and superscript where we refer to the rate of a single arbitrary SU on any channel and instead use R.(ii)Power: The power of the *k*th SU on the *i*th channel shall be denoted by $P_{k}^{i}$ for i=1…I and k=1…K. The power of an SU shall be measured in mW. The aim of optimization will be to minimize the total power of the entire network.(iii)Delay: This is the amount of time required to transfer all the bits of a packet data of size Ls through the channel. The transmission delay for the *k*th SU on the *i*th channel shall be denoted by $\tau_{k}^{i}$ for i=1…I and k=1…K. The transmission delay will be measured in Bps.(iv)Channel Selection: We assume that at any given instant, an SU is either ON a channel or OFF the channel. Thus, we use the binary variable γ to represent if a particular channel is selected by an SU or not:
(1)$$\gamma_{k}^{i}=\left\{\begin{array}{cc} 1\;: & \mathrm{if}\;\mathrm{SU}\;k\;\mathrm{selects}\;\mathrm{channel}\;i\;\forall \;k=1\ldots K,\;i=1\ldots I\\ 0\;: & \mathrm{Otherwise} \end{array}\right.$$

### 3.2. Primary User Model

The PU is assumed either to be ON a channel or OFF the channel. The time spent ON and OFF the channel is modeled as independent and identically distributed random variables with mean values $\mu_{on}$ and $\mu_{off}$, respectively [1]. The probability of a PU being ON the channel, $p_{on}$ is thus given by the following equation:(2)$$p_{on}=\frac{\mu_{on}}{\mu_{on}+\mu_{off}}$$

The probability of a PU being OFF the channel, $p_{off}$ is given by the following equation:(3)$$p_{off}=\frac{\mu_{off}}{\mu_{on}+\mu_{off}}$$

The fact that we adopt the overly spectrum approach means that the bigger the size of $p_{off},$ the more chances of having an SU on that particular channel.

### 3.3. Channel Sensing Model

For secondary users to begin transmitting, they need to first sense the spectrum for possible PU transmission on the channel. Thus, for channel sensing, we use compressive sensing techniques. This reduces the internal sensing power consumption problem since the sensing nodes are not mobile and may thus not necessarily be powered by batteries; it also increases spectrum efficiency since the cognitive radios do not do sensing and overcomes the hidden primary user problem and uncertainty brought about by shadowing and fading [2]. In order to model the channel, we need to first compute the probability of false alarm, which is the probability that a free channel is incorrectly sensed as having a PU on it, and the probability of mis-detection, which is the probability of incorrectly concluding that a channel is free, yet a PU exists on it. For external sensing, we adopt the model proposed by [34]. Since the external node does the sensing on behalf of the SUs, the SUs will be in wake up mode if and only if they have data to transmit, otherwise, they are in sleep mode which helps them save the energy which would be needed in listening to the external network. For energy detection, it is hypothesized that the signal at the external node at any time t∈[0,T] is given by Equation (Equation 4):(4)$$y\left(t\right)=\left\{\begin{array}{cc} x(t)+n(t)\; & :\;\mathrm{channel}\;\mathrm{busy},\;\;H_{1}\\ n(t)\; & :\;\mathrm{channel}\;\mathrm{idle},\;\;H_{0} \end{array}\right.$$
where y(t) is the signal received by the external node, $n\left(t\right)∼N\left(0,\sigma_{n}^{2}\right)$ is the Additive White Gaussian Noise (AWGN) described by a normal distribution with mean 0 and variance $\sigma_{n}^{2}$, x(t) is the transmitted PU signal, assumed to be Circularly Symmetric Complex Gaussian (CSCG) distributed with variance $\sigma_{x}^{2}$, $H_{0}$ and $H_{1}$ are the hypotheses that the PU is OFF the channel and ON the channel, respectively. Channel sensing is done by energy detection. The probability of false alarm $p_{f}$ and the probability of mis-detection $p_{m}$ are given as [1]:(5)$$p_{f}=Q\left(\frac{\lambda_{T}-N\sigma_{n}^{2}}{\sqrt{2N\sigma_{n}^{2}}}\right)$$
(6)$$p_{m}=1-Q\left(\frac{\lambda_{T}-N\left(\sigma_{n}^{2}+\sigma_{x}^{2}\right)}{\sqrt{2N(\sigma_{n}^{4}+\sigma_{x}^{4})}}\right)$$
where $\lambda_{T}$ is signal detection threshold, and $Q\left(.\right)$ is the right tail probability of the standard normal distribution. If $t_{s}$ is the sensing time and *W* the channel bandwidth, then the sample size *N* is defined as in the following equation:(7)$$N=2t_{s}W.$$

We adopt a decentralized dynamic spectrum access for SUs, where each SU senses for channels separately. We assume that an SU can only transmit one packet of data in every round. Whenever an SU has data packets to transmit, any of the following states may occur on a given channel. We adopt the states developed in [4]:(i)Detection: Under this state, the SU senses the presence of a PU on the channel. The probability of detection $p_{d}$ is given as
(8)$$p_{d}=p_{on}-p_{m},$$
where the probability of a PU being ON the channel $p_{on}$ is given by (Equation 2), and the mis-detection probability $p_{m}$ is given by (Equation 6).(ii)Mis-detection: Under this state, the channel is occupied by a PU but the SU incorrectly senses it to be a free channel. The probability of mis-detection is given by Equation (Equation 6).(iii)False Alarm: The channel is free and would be available for transmission usage by the SU but the SU misreads the channel to be occupied. This means that this particular SU cannot transmit on this channel at that instant. The probability of false alarm $p_{f}$ is given by Equation (Equation 5).(iv)SU channel co-existence: If the channel bandwidth *W* is high enough for two or more SUs to transmit on a single channel, then transmission can be done for all SUs that select that particular channel. The probability that two or more SUs select one channel to transmit on $p_{co}$ is given by the following equation:
(9)$$p_{co}=\left(p_{off}-p_{f}\right)\left[1-{\left(1-\frac{1}{I}\left(1+\frac{p_{off}}{p_{on}}\right)\right)}^{K-1}\right].$$(v)Collision: Since a PU is the first priority user of a channel, if a PU appears and needs to transmit on a channel which is already occupied by a SU, collision happens and the SU would be interrupted and have to vacate the channel for the PU. The probability of collision, $p_{col}$ is given by
(10)$$p_{col}=\left(1-e^{-\frac{Ls}{\mu_{off}R}}\right)\left(p_{off}-p_{f}-p_{co}\right).$$(vi)Successful Transmission: This is the ideal good situation for an SU. Under this state, an SU correctly senses a free channel and successfully transits on the channel without failure. For a successful transmission by an SU to occur, first the channel should be in OFF mode at the time of sensing, then false alarm should not occur, there should be no PU arriving to cause collision and there should not be coexistence state. The probability of successful transmission, $p_{suc}$ is given by
(11)$$p_{suc}=p_{off}-p_{f}-p_{co}-p_{col}.$$

## 4. Problem Formulation

In this section, we formulate our optimization model. We discuss the QoS constraints before formulating the model employed.

### 4.1. Quality of Service (QoS) Constraints

Here, we explain the two QoS constraints that are considered in this work i.e., PU interference and Bit Error Rate (BER) constraints. Both constraints follow the set up developed in [4].

#### 4.1.1. Primary User Interference

Interference of PU transmission occurs as a result of interruption from an SU. Interference of PU may occur in case of mis-detection from an SU or in case of collision i.e., when a PU finds an SU on its channel. Thus, PU interference should be a function of both $p_{m}$ and $p_{col}$. Interference is defined as the ratio of average PU interference time to the average ON time $\mu_{on}$ and is given by
(12)$$I_{PU}=\frac{Ls}{R}\left[\frac{p_{m}\left(p_{on}+p_{off}e^{-\frac{Ls}{R\mu_{on}}}\right)+p_{col}p_{on}}{p_{m}+p_{col}}\right]\frac{1}{\mu_{on}}.$$

#### 4.1.2. Bit Error Rate (Ber)

We consider a noisy channel such that the Bit Error Rate (BER) is the percentage of bits of a data packet that are corrupted due to either noise alone or noise together with interference. Let $S_{N}$ be the Signal-to-Noise-Ratio on a noisy channel and $S_{I}$ be the Signal-to-Noise-Ratio on a noisy channel with interference (Signal-to-Interference-Noise-Ratio (SINR)). Considering a Rayleigh fading channel with a Rayleigh fading component of ρ, the probabilities of $S_{N}$ and $S_{I}$ follow a Chi-square distribution. The expected values of $S_{N}$ and $S_{I}$ are given by
(13)$${\hat{S}}_{N}=\frac{E\left(\rho^{2}\right)P_{s}}{N_{0}R}$$
and
(14)$${\hat{S}}_{I}=\frac{E\left(\rho^{2}\right)P_{s}}{(N_{0}+P_{p})R}$$
respectively. $P_{s}$ is the received SU power, $P_{p}$ the received PU power and $N_{0}$ is the noise spectral density. The received power $P_{s}$ and $P_{p}$ are estimated using the Log-normal shadowing model with variance due to shadowing $\sigma_{s}^{2},$ path-loss exponent *l*, reference distance of 1 m, distance between two CRSN nodes $d_{s},$ distance between CRSN node and PU $d_{p}$ and a signal of wavelength λ operating at a frequency *f*.

Using Frequency Shift Keying (FSK) modulation on a Rayleigh fading channel, the BER of an SU is given by [4]:(15)$$P_{BER}=\frac{\left(2+{\hat{S}}_{I}\right)+\Sigma \left({\hat{S}}_{N}-{\hat{S}}_{I}\right)}{\left(2+{\hat{S}}_{I}\right)\left(2+{\hat{S}}_{N}\right)},$$
where
(16)$$\Sigma =\frac{pm\left(p_{on}+p_{off}e^{-\frac{Ls}{R\mu_{on}}}\right)+p_{co}+p_{on}p_{col}}{1-(p_{d}+p_{f})}.$$

### 4.2. Optimization Model

In this sub section, we formulate a multi-objective optimization model with three objective functions, which are total network power, total network rate and total network delay. The aim of the optimization problem is to jointly maximize total network rate, minimize total network power and minimize total network delay, subject to a number of constraints. Since the channel selection variable $\gamma_{k}^{i}$ is binary, we thus end up with a Mixed Integer Non-Linear Programming (MINP) Problem:
(17a)$$\begin{array}{cc} \max\; & \sum_{i=1}^{I}\sum_{k=1}^{K}\gamma_{k}^{i}R_{k}^{i} \end{array}$$
(17b)$$\begin{array}{cc} \min\; & \sum_{i=1}^{I}\sum_{k=1}^{K}\gamma_{k}^{i}P_{k}^{i} \end{array}$$
(17c)$$\begin{array}{cc} \min\; & \sum_{i=1}^{I}\sum_{k=1}^{K}\gamma_{k}^{i}\tau_{k}^{i} \end{array}$$
(17d)$$\begin{array}{cc} s.t.\; & 0\leq P_{k}^{i}\leq {\hat{P}}_{k}^{i}\;\forall k=1\ldots K,\;i=1\ldots I \end{array}$$
(17e)$$\begin{array}{cc} \; & 0\leq \tau_{k}^{i}\leq {\hat{\tau }}_{k}^{i}\;\forall k=1\ldots K,\;i=1\ldots I \end{array}$$
(17f)$$\begin{array}{cc} \; & 1\leq R_{k}^{i}\leq W\log_{2}\left(1+\frac{P_{k}^{i}}{WN_{0}}\right)\;\forall k=1\ldots K,\;i=1\ldots I \end{array}$$
(17g)$$\begin{array}{cc} \; & \gamma_{k}^{i}=\left\{0,1\right\},\;\forall k=1\ldots K,i=1\ldots I \end{array}$$
(17h)$$\begin{array}{cc} \; & \sum_{k=1}^{K}P_{k}^{i}\gamma_{k}^{i}\leq {\bar{P}}^{i},\;\forall i=1\ldots I \end{array}$$
(17i)$$\begin{array}{cc} \; & \sum_{k=1}^{K}R_{k}^{i}\gamma_{k}^{i}\leq {\bar{R}}^{i},\;\forall i=1\ldots I \end{array}$$
(17j)$$\begin{array}{cc} \; & \tau_{k}^{i}=\frac{Ls}{R_{k}^{i}},\;\forall k=1\ldots K,i=1\ldots I \end{array}$$
(17k)$$\begin{array}{cc} \; & \sum_{k=1}^{K}I_{PU}^{i}\leq I_{max}^{i},\;\forall i=1\ldots I \end{array}$$
(17l)$$\begin{array}{cc} \; & P_{BER}^{i}\leq {BER}_{max}^{i},\;\forall k=1\ldots K,i=1\ldots I \end{array}$$
(17m)$$\begin{array}{cc} \; & \sum_{i=1}^{I}\gamma_{k}^{i}\geq 1,\;\forall k=1\ldots K. \end{array}$$

From the optimization problem (17), the total channel rate, total channel power and total channel delay are are given by Equation (17a–c), respectively. Constraints (17d) and (17e) represent the lower and upper bounds for power and transmission delay, respectively. The upper bounds for both power and transmission delay are both assumed to be constants. From constraint (17f), the rate is assumed to be bounded below by 1 and above by a function which is dependent upon the power. Constraint (17g) stands for the binary variable which has been defined in Equation (Equation 1). It is assumed that each channel has maximum allowable power and rate. From constraint (17h), the maximum allowable channel power for all users is ${\bar{P}}^{i}$ and from constraint (17i), the maximum allowable rate on channel *i* is given as ${\bar{R}}^{i}$. These upper bounds on total channel power and rate are restrictions which are normally brought about by the hardware capabilities. Constraint (17j) presents the transmission delay as a function of rate and the fixed packet data size Ls [35]. Constraints (17k) and (17l) are the QoS constraints that have been explained in Section 4.1.1 and Section 4.1.2, respectively. We assume in this model that an SU is allowed to transmit on one or more channels, if conditions permit. This is captured in constraint (17m).

The optimization problem (17) can be re-written as a single objective optimization by forming a weighted sum of the objective Equation (17a–c). Let $\lambda_{1}$, $\lambda_{2}$ and $\lambda_{3}$ be the weights inserted on (17a–c), respectively, such that $\lambda_{1}+\lambda_{2}+\lambda_{3}=1.$ This technique is called *linear scalarization method* or *weighted sum approach* [36,37].

Suppose that the objective Equation (17a–c) are written as
(18)$$F_{1}=\sum_{i=1}^{I}\sum_{k=1}^{K}\gamma_{k}^{i}R_{k}^{i}$$
(19)$$F_{2}=\sum_{i=1}^{I}\sum_{k=1}^{K}\gamma_{k}^{i}P_{k}^{i}$$
(20)$$F_{3}=\sum_{i=1}^{I}\sum_{k=1}^{K}\gamma_{k}^{i}\tau_{k}^{i},$$
then the multi-objective optimization problem (17) can be re-written as a single objective optimization problem:
(21a)$$\begin{array}{cc} \min\; & -\lambda_{1}\beta_{1}F_{1}+\lambda_{2}\beta_{2}F_{2}+\lambda_{3}\beta_{3}F_{3} \end{array}$$
(21b)$$\begin{array}{cc} s.t.\; & (17d),(17e),(17f),(17g),(17h),(17i),(17j),(17k),(17l),(17m). \end{array}$$

Here, $\beta_{1},\beta_{2}$ and $\beta_{3}$ are called normalization factors. The use of the normalization factors is to ensure that we get Pareto optimal solutions that are consistent with the weights $\lambda_{1},\lambda_{2}$ and $\lambda_{3}.$

It should be noted that for a fixed set of values $\lambda_{1}>0,\lambda_{2}>0$ and $\lambda_{3}>0$, the optimal solutions of (21) form a Pareto front of (17). This establishes the equivalence between the optimal solutions of (17) and (21). We therefore solve Equation (21) for the optimal rate, power, delay and channel allocation. The procedure that we use for calculating the normalization factors is the use of the difference of optimal values in the Nadir and Utopia points [36].

Let
(22)$$\begin{array}{cc} X_{\left[i\right]}^{*}= & \mathrm{argmin}\;\{F_{i}\left(x\right):x\;\mathrm{satisfies}\;\left(17d\right),\left(17e\right),\left(17f\right),\left(17g\right),\left(17h\right),\left(17i\right),\left(17j\right),\left(17k\right),\left(17l\right),\left(17m\right)\}\\ \; & \mathrm{for}\;i=1,2,3. \end{array}$$

Then
(23)$$U_{i}=F_{i}\left(X_{\left[i\right]}^{*}\right),\;\mathrm{for}\;i=1,2,3,$$
are called the Utopia points. The Nadir points are defined as in the equation below:(24)$$N_{i}=\max\left\{F_{i}\left(X_{\left[1\right]}^{*}\right),F_{i}\left(X_{\left[2\right]}^{*}\right),F_{i}\left(X_{\left[3\right]}^{*}\right)\right\},\;\mathrm{for}\;i=1,2,3.$$

The normalization factors are thus defined as
(25)$$\beta_{i}=\frac{1}{N_{i}-U_{i}},\;\mathrm{for}\;i=1,2,3.$$

The corresponding normalized functions for (Equation 18)–(Equation 20) are thus
(26)$$\frac{F_{i}-U_{i}}{N_{i}-U_{i}}\in \left[0,1\right]\;\mathrm{for}\;\;i=1,2,3.$$

For each selected set of values $\lambda_{1},\lambda_{2}$ and $\lambda_{3},$ the normalization factors are calculated after solving three separate single-objective optimization problems (Equation 22).

## 5. Numerical Results

In this section, we provide both the solution method used to solve model (21) and also the numerical results. The results that are obtained are used to demonstrate the effect of joint optimization of transmission power, transmission rate and transmission delay in a CRWSN.

We consider a network in with 3 PUs and 10 CRSN nodes. The distance between any two CRWSN nodes in the system is 9 m whereas the CRSN node to PU distance shall be 80 m. In our network model, we allow for fading due to shadowing, with a path-loss exponent of 3.5 [4]. The received PU power at a CRSN node receiver, power spectral density, antenna gains of receiver and transmitter of CRSN node are all provided in Table 1. The maximum power, maximum delay and maximum BER for each channel are also 100 mW, 160 ms and 0.01 respectively. The rest of the parameters used in the simulations for our network model are given in Table 1.

In order to solve the problem, we employ a branch-and-cut approach that has been developed and extensively studied in the literature, for example [38]. This technique is an exact global optimization method for solving non-convex and non-linear optimization problems. The technique involves developing polyhedral cutting planes and relaxations at each branch. The main idea of this technique is to recognize convexity in the functions of the optimization problem. Once a function is a known and recognizable convex function, then better relaxations and approximations of it can easily be made. For those functions that are not convex, either prior studied convex function compositions of it are used or even a user is allowed to supply their own better estimators. It has been proved in [38] and other works that the polyhedral cutting planes generated by this method are much tighter than any outer approximations of the original functions. This makes this technique much faster than other outer approximation methods. These polyhedral relaxations are efficiently handled by most state-of-the-art linear programming solvers. For our case, we use CPLEX 12.5 as the linear programming solver. This technique has been developed into a solver called BARON which is accessible from https://www.minlp.com. In order to run simulations of our model, we use MATLAB together with the BARON solver.

We also explore further the effect of using different scalarization factors $\lambda_{1},\lambda_{2}$ and $\lambda_{3}$ in problem (21). It should be noted that the size of the weights $\lambda_{1},\lambda_{2}$ and $\lambda_{3}$ is an indication of the relative weight or importance put on total rate, total power and total transmission delay, respectively, during optimization. We thus come up with four different scenarios or cases explained in terms of the relative importance or weight put on all the three objective Equations (18)–(20) of the optimization problem (17). The first is when equal weight ($\lambda_{1}=\lambda_{2}=\lambda_{3}=\frac{1}{3}$), is put on all the three objective functions of (17), which we call case 1. The second, third and forth cases are when more weight is put on transmission rate, transmission power and transmission delay, respectively. These cases are summarized in Table 2.

### 5.1. Maximum Bit Error Rate and Maximum Interference

We study the effect of changing the maximum Bit Error Rate, ${\mathrm{BER}}_{\max}$ and maximum interference $I_{max}$ on the optimal values of total rate, total power and total transmission delay for the entire network.

In order to study the effect of varying the maximum interference, $I_{max}$ on the total rate, total delay and total power, we fix the packet size to Ls=300 Bytes and use all the other parameters as provided in Table 1. The results of the simulations are given in Figure 1.

We also set the packet size to Ls=300 Bytes and vary the maximum BER (${\mathrm{BER}}_{\max}$), in order to study the effect of ${\mathrm{BER}}_{\max}$ on the total power, total rate and total delay. The results of this are summarized in Figure 2.

From Figure 1a,b, it can be observed that both the total rate and total power are decreasing functions of $I_{max}$ for values of $I_{max}$ less than 0.18 and 0.1, respectively. The curves of total rate and total power start to get flat as $I_{max}$ increases beyond 0.18 and 0.1, respectively. The total delay curves also start to become relatively flat after a maximum interference value of 0.18. Thus, we can conclude that for a packet size of 300 Bytes and the parameters in Table 1, an interference of 0.18 is a good threshold value for PUs. This is because both graphs are generally flat for the values $I_{max}\geq 0.18.$ An increase in PU interference generally leads to a decrease in the rate of transmission and the power of the SUs. However, an increase in PU interference will generally lead to an increase in the time taken by an SU to transmit on the channel, hence the increase in transmission delay. These behaviors are shown in the simulation results in Figure 1. The behavior can be attributed to the fact that the SUs are not the first priority users of the channels and have to vacate the channels every time a PU comes on the channel.

It should also be noted that case 3 (more weight on power) leads to the least amount of power and rate, and also the highest amount of delay. On the other hand, case 4 (more weight on power) leads to the lowest transmission delay and the highest power and rate. The case of equal weights for all the objective functions, case 1, gives optimal values between case 3 and case 4.

From Figure 2a,b, the total rate and total power are an increasing function of maximum BER, until a threshold value of ${\mathrm{BER}}_{\max}=0.01$ is reached. On the other hand, the total delay is a decreasing function of maximum BER until a threshold value of ${\mathrm{BER}}_{\max}=0.01$ is reached, as seen from Figure 2c. After the threshold value of ${\mathrm{BER}}_{\max}=0.01$, is reached, the curves for total delay, total rate and total power become flat. This shows that the threshold value for ${\mathrm{BER}}_{\max}$ for our channel is 0.01. Before reaching the threshold ${\mathrm{BER}}_{\max}=0.01$, the total power and total rate are an increasing function of maximum BER, and the total transmission delay is a decreasing function of maximum BER. This is because of the fact that allowing for a higher percentage of corrupted bits leads to an increase in total transmission rate and power, and at the same time leads to less transmission time. Using case 1 gives results that are generally in-between (cases 2, 4) and case 3. Optimizing using case 3 gives the least power and rate and also the highest delay, whereas using (cases 2, 4) gives the highest power and rate, and the lowest transmission delay.

### 5.2. Packet Size

In order to study the effect of packet size on optimal total transmission rate, total transmission power and total transmission delay, we vary the packet size from 100 Bytes to 1000 Bytes [1]. The results for the simulations using parameters in Table 1 and the cases in Table 2, are given in Figure 3.

From Figure 3a–c, it can be observed that generally, an increase in packet size leads to an increase in the total transmission rate, total power and total delay, for all the four cases of Table 2. This is attributed to the fact that as packet size increases, the chances of interference with PUs also increases and also an increase in packets being corrupted due to noise becomes more likely. This therefore leads to an increase in transmission delay and also an increase in total system power and rate, due to more re-transmissions as a result of failures or collisions with PUs.

Putting more weight on rate (case 4) yields the highest rate, power and delay, as packet size is increased.

As observed from Figure 4a–c, using case 4 leads to a higher total power and total rate of the system, but at the expense of a higher transmission delay. However, using case 3 leads to the lowest power and rate, and a less transmission delay than for case 4.

### 5.3. Network Size

Here we study the effect of increasing the network size, on the optimal values of rate, power and delay. Network size here means the number of CRSN nodes. We thus vary the number of SUs on the channel from 5 to 30 and report the results in Figure 4.

As seen from Figure 4a–c, the total transmission power, total transmission rate and total transmission delay are increasing functions of number of CRSN nodes. This trend can be observed in all the four cases. This is attributed to the fact that increasing the number of SUs on the network leads to more data packets being transferable which in turn leads to increased total rate, total delay and total power.

The use of case 3 leads to less power and rate, with a higher transmission delay. The use of case 4 however leads to the highest power and rate, with a relatively smaller delay.

### 5.4. Primary User Interference

The PU interference is given by Equation (Equation 12). The values of total channel interference for each channel are not necessarily the maximum values $I_{max}^{i}$ for each channel. In this section, we look at the total channel interference values for the different network sizes and packet size values. The results are given in Figure 5.

It can be observed from Figure 5a,b that both network size and packet size increase with total interference. As the network size (number of SUs) increases, the chances of interference with PUs also increases. This explains the general increase of PU interference with network size. In addition, an increase in packet size leads to an increase in SU activity, which in turn leads to an increased chance of interference with a PU on the channel.

### 5.5. Bit Error Rate

In this sub-section, we study the effect of packet size and network size on the BER. The BER of each user is given by Equation (Equation 15). The total BER of the network is calculated for each packet size and network size and the results are summarized in Figure 6.

The total BER is in general an increasing function of packet size, as seen from Figure 6a,b. The reason for the increase is that an increase in packet size with all other system capabilities kept constant, leads to an increase in the possibility of data getting more corrupted due to increased collisions among the channel users. The general non-smoothness of the graphs of BER is attributed to the fact that BER is not constrained as a total channel BER but rather individual BER for each user on a channel. The total BER also increases with increasing network size. This is because the more the number of users on the network increases, the higher the chances of collisions and interference. This in turn leads to an increase in a higher possibility of corrupted packets.

## 6. Conclusions and Recommendation

We have jointly optimized three variables namely; transmission rate, transmission power and transmission delay needed by the SUs to transmit in a CRSN amidst a number of constraints including interference, BER, boundary constraints on the variables, channel power and rate constraints. The resulting channel selection problem is a multi-objective mixed integer non-linear programming problem. The resulting problem has been solved using a branch-and-cut polyhedral approach using MATLAB together with the state-of-the-art BARON software. We have analyzed the optimal values of the problem i.e., the total transmission rate, total transmission power and total transmission delay, with changing values of packet size, network size, interference and BER. Our set up also puts into consideration the effect of shadowing during transmission.

We have observed that in general, an increase in network size and packet size leads to an increase in total rate, total power, total transmission delay, total interference and total BER. It is worth noting that we do not compare with any other work because other scholars have only optimized one or two variables but not all the three as we have done in this study.

For all the other cases, the rate, delay and power generally increase with increasing size of network. It has also been observed that an increase in packet size generally leads to an increase in total rate, power and delay. The choice of cases 1, 2, 3 and 4 generally leads to different optimal values, albeit in the same direction. It is therefore the choice of the system manager/user to decide the choice of case, depending on where more weight is needed.

For future work, the authors are already trying to devise a heuristic technique to solve the problem for a larger set of parameter values and network sizes. In addition, further modifications to the studied problem are being considered to incorporate ideas like SU prioritization, more constraints, etc.

## Figures and Tables

**Figure 1 sensors-20-04907-f001:**
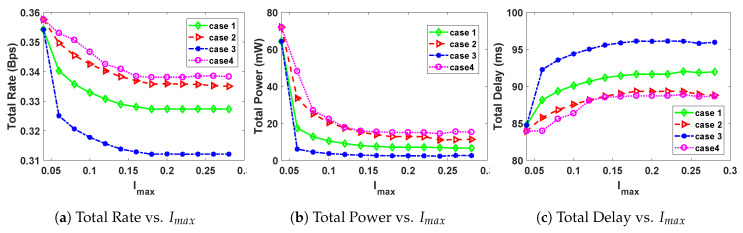
Effect of changing maximum interference $I_{max}$, on the optimal rate, power and delay.

**Figure 2 sensors-20-04907-f002:**
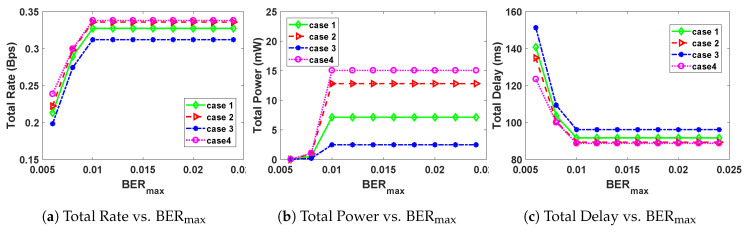
Effect of changing maximum interference ${\mathrm{BER}}_{\max}$, on the optimal rate, power and delay.

**Figure 3 sensors-20-04907-f003:**
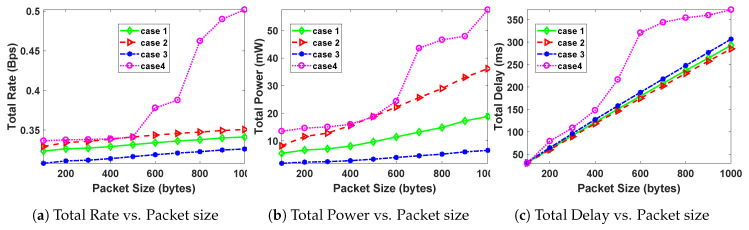
Effect of packet size on the optimal rate, power and delay.

**Figure 4 sensors-20-04907-f004:**
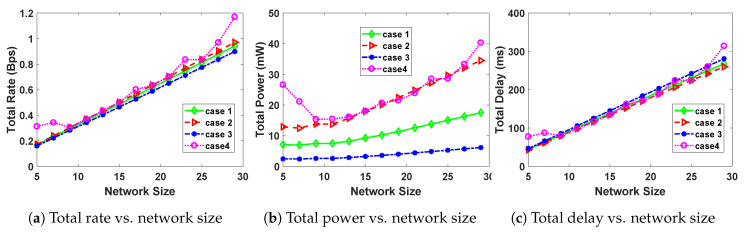
The effect of changing the number of Cognitive Radio Sensor Network (CRSN) nodes on the optimal solution of (17).

**Figure 5 sensors-20-04907-f005:**
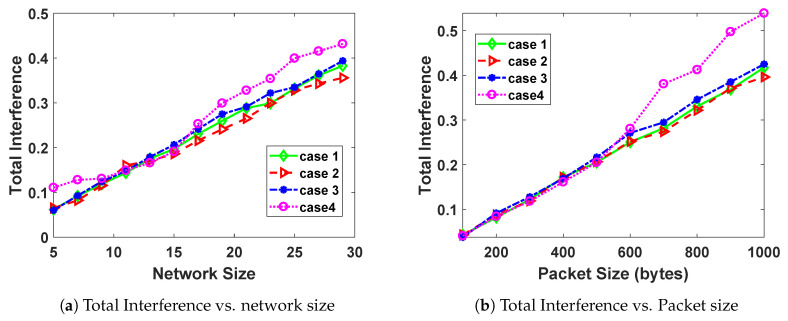
Effect of network size and packet size on the total interference.

**Figure 6 sensors-20-04907-f006:**
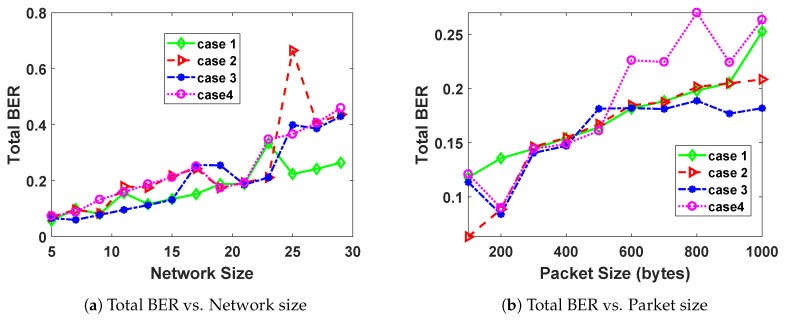
The effect of increasing network size and packet size, on the total Bit Error Rate (BER) of the network.

**Table 1 sensors-20-04907-t001:** Parameter values used in the simulations.

Parameter	Meaning	Numerical Value
$p_{on}$	Probability that the PU is ON	0.3
$I_{max}$	Maximum Interference	0.18
${BER}_{max}$	Maximum BER	0.01
*I*	Number of channels	3
*K*	Number of CRSN Nodes	10
$p_{f}$	Probability of false Alarm	0.1
*f*	Frequency of transmission	1 GHz
$N_{0}$	Noise spectral density	$1\times 10^{-10}$
*W*	Band Width	1 MHz
$\bar{P}$	Maximum Power on each Channel	100 mW
$\hat{\tau }$	Maximum Delay for each user	160 ms
$\mu_{on}$	Average times of PUs being ON channel	[0.60,0.64,0.86]/s
$\sigma_{s}$	Standard deviation due to shadowing	6 dB
$P_{u}$	PU received power	10 dB
$G_{t}$	Transmit Antenna Gain	2 dBi
$G_{r}$	Receive Antenna Gain	0 dBi
$d_{s}$	Distance between two SUs	9 m
$d_{p}$	Distance between CSN node and PU	80 m
*l*	Path-loss Exponent	3.5

**Table 2 sensors-20-04907-t002:** Different cases used in simulations.

	$\lambda_{1}$	$\lambda_{2}$	$\lambda_{3}$
case 1	$\frac{1}{3}$	$\frac{1}{3}$	$\frac{1}{3}$
case 2	0.2	0.2	0.6
case 3	0.2	0.6	0.2
case 4	0.6	0.2	0.2

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
