# Peer review of "A Joint Power, Delay and Rate Optimization Model for Secondary Users in Cognitive Radio Sensor Networks"

_sensors, 2020, doi:10.3390/s20174907_

Round 1
Reviewer 1 Report
The paper optimizes transmission rate, transmission power, and transmission delay for cognitive radio sensor networks. In general, the paper is well written and the topic is interesting. The references used in the paper also look right.
If possible, please make plots larger. They are difficult to watch.
Author Response
The paper optimizes transmission rate, transmission power, and transmission delay for cognitive radio sensor networks. In general, the paper is well written and the topic is interesting. The references used in the paper also look right.
We thank the this reviewer for these compliments.
If possible, please make plots larger. They are difficult to watch.
The plots have now been increased to the largest possible sizes in their lines. We hope that they are more visible now. If not, we can still have a maximum of two plots per line and push the third to the next line. It will only have one effect of increasing number of pages.
Reviewer 2 Report
The paper “A Joint Power, Delay and Rate optimization Model for Secondary Users in Cognitive Radio Sensor Networks” presents an attempt to find the optimal size of packages in a Cognitive Radio Sensor Networks. The aim is to minimize the power consumption and communication delay, and to maximize the transmission rate simultaneously using a multi-objective optimization model.
The paper starts with problem statement in context of state of the art in the domain. However, the literature review presented here is difficult to read. The authors should avoid very long sentences and to mix many different ideas in a single paragraph. A better structure can definitively increase the clarity.
The next sections present in detail the proposed optimization technique. Some flaws should be corrected as equations that are not numbered. (e.g. in [250], [254] etc.), and some terms that are not explained (e.g. Q in (3) and (4), p0n in [254] etc.).
The Numerical Results section presents an attempt to validate the approach by solving the optimization model and computing the impact on some QoS parameters defined here as Interferences and BER. However, the authors do not include any description of the network architecture used for simulation. Many flows can be observed on interpretation of the results. Figure 1 is hard to be explained. What is the meaning of “Toal Delay” (should be typo)? It is not clear what unit is used on the vertical axes to represent the magnitude of delay, power and rate simultaneously (ms, mW, bps or the size?). Moreover, it is not clear how the conclusion in [328] is drawn from the figure. The authors do not explain how the cases presented in Table 2 are used in the selection of the optimal packet size, as Figure 1 covers just the first case. The results presented in the figures 2, 3, 4, and 5 are not explained. The authors do not investigate the meaning of the obtained values. E.g. why the minimum power is obtained for 10 nodes in figure 3(b)?
The last section presents conclusions and future work. However, some conclusions are not easy to be explained since the authors do not include any proof as [405] “This is because the resulting problem is NP-hard.”
Author Response
Extensive editing of English language and style required
The entire document has been read through again to ensure that there are no grammatical errors or excessively long sentences. Although the reviewer did not specify the exact sentences requiring editing, we still tried to change a few lines in the introduction to ensure the intended meaning is not ambiguous.
The paper “A Joint Power, Delay and Rate Optimization Model for Secondary Users in Cognitive Radio Sensor Networks” presents an attempt to find the optimal size of packages in a Cognitive Radio Sensor Networks. The aim is to minimize the power consumption and communication delay, and to maximize the transmission rate simultaneously using a multi-objective optimization model.
This is a general statement from the reviewer about the article.
The paper starts with problem statement in context of state of the art in the domain. However, the literature review presented here is difficult to read. The authors should avoid very long sentences and to mix many different ideas in a single paragraph. A better structure can definitively increase the clarity.
The entire literature review section has been re-written to ensure it is much easier to read. We have avoided long sentences in the new write-up. There are now three paragraphs in total under the literature review section. The first paragraph handles literature on network optimization problems (mainly single objective problems), the second paragraph handles multi-objective optimization literatre and the third paragraph is devoted to general algorithms and solution techniques.
The next sections present in detail the proposed optimization technique. Some flaws should be corrected as equations that are not numbered. (e.g. in [250], [254] etc.), and some terms that are not explained (e.g. Q in (3) and (4), p0n in [254] etc.).
Equations on the lines [250], [254] are now all numbered. In fact all the equations in the entire paper are now numbered and referenced. The Q in (3) has now been explained immediately after equation (6). The pon has now been referenced with equation (2). The same has been done with pm, in equation (6), and all the other terms in the entire article.
The Numerical Results section presents an attempt to validate the approach by solving the optimization model and computing the impact on some QoS parameters defined here as Interferences and BER. However, the authors do not include any description of the network architecture used for simulation. Many flows can be observed on interpretation of the results. Figure 1 is hard to be explained. What is the meaning of “Toal Delay” (should be typo)? It is not clear what unit is used on the vertical axes to represent the magnitude of delay, power and rate simultaneously (ms, mW, bps or the size?). Moreover, it is not clear how the conclusion in [328] is drawn from the figure. The authors do not explain how the cases presented in Table 2 are used in the selection of the optimal packet size, as Figure 1 covers just the first case. The results presented in the figures 2, 3, 4, and 5 are not explained. The authors do not investigate the meaning of the obtained values. E.g. why the minimum power is obtained for 10 nodes in figure 3(b)?
The network architecture for simulation has been explained in the second paragraph of section 5. “Toal Delay” was a typo that was meant to be “Total Delay”. Figure 1 from the earlier version has been replaced with both Figures 1 and 2. This has been done to ensure uniformity with all the other remaining figures in the section. All figures now consider the four cases that are being studied. All the figures have been re-done with the units of the vertical axes included…that is ms for delay, Bps for rate and mW for power. The conclusion [328] has been explained further in the fourth paragraph of section 5.1. Moreover, the two added Figures 1 and 2 help to explain it further. An extra paragraph, on top of what was there for Figures 2,3,4 and 5 has been added. The line on minimum power for 10 nodes has been entirely removed. This is because we realized that such a behavior could have been caused by a number of other factors like shadowing, increased non-linearity of the model at that case, etc. Thus we opted to leave out such a conclusion. We sincerely thank the reviewer for helping us to point out that issue.
The last section presents conclusions and future work. However, some conclusions are not easy to be explained since the authors do not include any proof as [405] “This is because the resulting problem is NP-hard.”
The authors are already in the process of writing another article for a heuristic solution, where statement [405] will be proved. The statement has been removed to avoid causing any confusion.
Reviewer 3 Report
The main objective of the presented work was to demonstrate an optimization model for Secondary Users in Cognitive Radio Sensor Networks taking into account jointly the power, power, and transmission rate, under constraints of Bit Error Rate (BER), interference and other channel limitations (effect of shadowing during transmission).
The proposed strategy for solving the problem was based on a branch-and-cut polyhedral approach and next verified within simulations by means of MATLAB and dedicated BARON software.
The Introduction and Related Literature sections provide a comprehensive description of the context of the work.
The network mode and the problem formulation were shown in sufficient details.
The numerical simulations proved that the presented three-variable model creates an effective tool for analyzing operation of Cognitive Radio Sensor Networks.
English style and the overall presentation are good.
One minor remark: the structure of the 3.1 section is odd; it is unclear why so many short subsections including only definitions are introduced. For better clarity of presentation I would not advise to not split it further into subsections 3.1.1 - 3.1.4, but instead merge then within section 3.1, so that all the four variables (Rate, Power, Transmission delay, and Channel selection) were described in details within it.
Author Response
The main objective of the presented work was to demonstrate an optimization model for Secondary Users in Cognitive Radio Sensor Networks taking into account jointly the power, power, and transmission rate, under constraints of Bit Error Rate (BER), interference and other channel limitations (effect of shadowing during transmission).
The proposed strategy for solving the problem was based on a branch-and-cut polyhedral approach and next verified within simulations by means of MATLAB and dedicated BARON software.
This is a very deep understanding of our problem. The authors really appreciated reading this.
The Introduction and Related Literature sections provide a comprehensive description of the context of the work.
The network mode and the problem formulation were shown in sufficient details.
The numerical simulations proved that the presented three-variable model creates an effective tool for analyzing operation of Cognitive Radio Sensor Networks.
English style and the overall presentation are good.
We very much appreciate the compliments given to us by the reviewer.
One minor remark: the structure of the 3.1 section is odd; it is unclear why so many short subsections including only definitions are introduced. For better clarity of presentation, I would not advise to not split it further into subsections 3.1.1 - 3.1.4, but instead merge then within section 3.1, so that all the four variables (Rate, Power, Transmission delay, and Channel selection) were described in details within it.
All the subsections under section 3.1 have now been merged into one section 3.1. The four variables are instead listed with numbers (i),…,(iv).
Round 2
Reviewer 2 Report
The new version of the manuscript “A Joint Power, Delay and Rate optimization Model for Secondary Users in Cognitive Radio Sensor Networks” represents an improvement on what was initially reviewed. The authors success to correct most of the signaled errors. The clarity of the literature review increases significantly. The model is a slightly better explained. Some confusing figures have been removed, the rest where improved and better explained. In my opinion, this version of the manuscript can be considered for publication.